# Role of intrapulmonary lymph nodes in patients with NSCLC and visceral pleural invasion. The VPI 1314 multicenter registry study protocol

**Fabrizio Minervini** [1]*, **Peter Kestenholz**[1], **Pietro Bertoglio** [2], **Allen Li**[3], **Henning Nilius**[4]

**1** Division of Thoracic Surgery, Cantonal Hospital Lucerne, Lucerne, Switzerland, **2** Division of Thoracic Surgery, IRCSS Azienda Ospedaliero-Universitaria, Bologna, Italy, **3** The Ottawa Hospital Research Institute, The Ottawa Hospital, Ottawa, Ontario, Canada, **4** Department of Clinical Chemistry, Inselspital, Bern University Hospital, Bern, Switzerland

* fabriziominervini@hotmail.com

**Data Availability Statement:** No datasets were generated or analysed during the current study. All relevant data from this study will be made available upon study completion.

## Abstract

### Background

In the lung cancer classification (TNM), the involvement of thoracic lymph nodes is relevant from a diagnostic and prognostic point of view. Even if imaging modality could help in selecting patients who should undergo surgery, a systematic lymph node dissection during lung surgery is mandatory to identify the subgroup of patients who can benefit from an adjuvant treatment.

### Methods

Patients undergoing elective lobectomy/bilobectomy/segmentectomy) for non-small cell lung cancer and lymphadenectomy with lymph nodes station 10-11-12-13-14 sampling that meet the inclusion and exclusion criteria will be recorded in a multicenter prospective database. The overall incidence of N1 patients (subclassified in: Hilar Lymph nodes, Lobar Lymph nodes and Sublobar Lymph nodes) will be examined as well as the incidence of visceral pleural invasion.

### Discussion

The aim of this multicenter prospective study is to evaluate the incidence of intrapulmonary lymph nodes metastases and the possible relation with visceral pleural invasion. Identifying patients with lymph node station 13 and 14 metastases and/or a link between visceral pleural invasion and presence of micro/macro metastases in intrapulmonary lymph nodes may have an impact on decision-making process.

### Trial registration

ClinicalTrials.gov ID: NCT05596578.

**Funding:** The author(s) received no specific funding for this work.

**Competing interests:** The authors have declared that no competing interests exist.

## Introduction

Lung cancer is the leading cause of cancer related death worldwide in men and women with non-small cell lung cancer (NSCLC) accounting for about 75% of cases [1]. Lung cancer classification depends on the extension of the tumor (T), involvement of lymph nodes (N) and presence of metastases (M). Lymph node staging has not only a diagnostic value but also a prognostic significance and is crucial towards establishing the optimal treatment strategy in patients [2]. Computed tomography (CT) of the chest is the most commonly used imaging technique to assess the primary tumor and for intra thoracic metastases. However, CT imaging has several limitations when used as the single imaging modality (sensitivity and specificity of 55% and 81%, respectively). PET imaging (sensitivity and specificity of 84% and 89% respectively, when evaluating for mediastinal metastasis), especially when combined with CT, plays an eminent role in the evaluation of intra-thoracic and extra thoracic metastases and is, therefore, recommended preoperatively for most patients suspected of having lung cancer [3]. Histological confirmation of mediastinal lymph nodes involvement detected in PET is a key point in order to establish the appropriate treatment of lung cancer. Despite the importance of an accurate nodal staging being well recognized, an international consensus regarding the extent of intra-operative mediastinal lymph node assessment is currently lacking. Current guidelines from The European Society of Thoracic Surgeons (ESTS) have recommended to perform a systematic dissection of the mediastinal lymph nodes during lung surgery dissecting at least six lymph nodes [4]. Concurrently, The American College of Surgeons' Commission on Cancer (ACS-CoC) revised the definition of good-quality curative-intent lung cancer surgery in 2020 requiring, specifically, the inclusion of a minimum of three named and/or numbered anatomic lymph node stations as well as at least one hilar or intrapulmonary lymph node [5].

However, the intrapulmonary lymph nodes (stations 13 and 14) are not routinely examined and it remains unknown whether dissecting them is necessary for accurate staging and prognostication. With the emergence of new and effective adjuvant treatment options, including immunotherapy and targeted agents, peripheral lymph node sampling could potentially be important.

Moreover, the prognostic significance of the visceral pleural invasion is controversial. Some studies showed a negative impact on overall survival (OS) and disease-free survival (DFS) in patients with histologic proved visceral pleura invasion (VPI, defined as invasion beyond the elastic layer—>PL1, including invasion to the visceral pleural surface—>PL2) [6–8].

The mechanism to explain this negative effect is not fully understood. Given that the visceral pleura on the pulmonary surface is very rich in lymphatic vessels and enters the parenchyma to connect with the bronchial lymph vessels to drain into various hilar nodes, we assume that the worse OS and DFS observed in these patients could be explained with the presence of metastatic lymph nodes (Station 13–14) that are not routinely examined.

## Material and methods

This is a prospective, multicenter study based on ad-hoc created prospectively database. Data will be collected through chart reviews from patient electronic medical record and institutional databases to identify patients who underwent elective lobectomy/bilobectomy/segmentectomy for NSCLC and lymphadenectomy with lymph nodes station 10-11-12-13-14 sampling.

N1 station location will be classified into 3 groups:

Hilar (10,11)

Lobar (12)

Sub-lobar (13–14)

| | STUDY PERIOD | | | | | | |
|---|---|---|---|---|---|---|---|
| | Enrolment | Post-op assessment | Follow up | | | | Close-out |
| TIMEPOINT | $t_0$ | $t_{0a}$ | $t_1$ (1Year post-op) | $t_2$ (2 Years post-op) | $t_3$ (3 Years post-op) | $t_4$ (4 Years post-op) | $t_5$ (5 Years post-op) |
| ENROLMENT: | | | | | | | |
| Eligibility screen | X | | | | | | |
| Informed consent | X | | | | | | |
| ASSESSMENTS: | | | | | | | |
| [baseline variables] | X | | | | | | |
| [primary outcome variables: N1 incidence, visceral pleural invasion incidence] | | X | | | | | |
| [secondary outcome variables: overall Survival, Disease free survival] | | | X | X | X | X | X |

**Fig 1. Schedule of assessments and evaluations.**

Data collected will be disseminated in an aggregate form after final collection to assure all participants remain anonymous (Fig 1). This protocol was approved by the Ethic Committee Nord-west and central Switzerland on December, 21, 2022 (Project number 2022–02217). Any change of the actual version of the protocol will be notified to the Ethic Committee with an appropriate written amendment.

The trial has been registered in clinicaltrials.gov (ID: NCT05596578)

The information recorded for analysis will only include indirect identifiers such as age, gender and hospital length of stay to minimize the risk of data breach.

## Primary outcomes

Overall incidence of N1 pathological lymph nodes (Hilar 10/11, Lobar 12, Sublobar 13/14)

Incidence of N1 pathological lymph nodes (Hilar 10/11, Lobar 12, Sublobar 13/14) in patients with pathological evidence of visceral pleural invasion.

## Secondary outcomes

Overall survival (1-3-5 years)

Disease free-survival (1-3-5 years)
Tumor recurrence (pattern: local, regional, distance)

## Inclusion criteria

Anatomical resection for NSCLC <3 cm (lobectomy, bilobectomy, segmentectomy) with radical mediastinal lymphadenectomy (at least 6 lymph nodes from 3 mediastinal nodal stations)
Samples from the intrapulmonary stations 12, 13, and 14 lymph nodes
Resection of lymph nodes station 10 and 11 during hilar separation
R0 resection

## Exclusion criteria

Prior or synchronous lung cancer
pN2
Pneumonectomy
R1/R2 resection
M1
Neo-adjuvant treatment

## Variables to be collected

Age
Sex (M/F/Other)
Cardiac comorbidity (Y/N)
Pulmonary comorbidity (Y/N)
Pre-op FEV1 (L,%)
Pre-op DLCO (%)
Pre-op PET (Y/N)
Pre-op mediastinal Staging (Y/N) if Y (EBUS/EUS, Mediastinoscopy, others + biopsied stations level)
Date of Surgery
Surgical access (Open, VATS, Robotic)
Site (RUL, MLL, RLL, LUL, LLL)
Type of resection (Wedge, Segmentectomy, Lobectomy, Bilobectomy, Pneumonectomy)
Length of stay
Tumor Size
Histology
Extent of lymphadenectomy
cTNM
pTNM
PL status
Metastatic Lymph nodes N1 (10, 11, 12, 13/14)
PDL1 (%)
EGFR Mutation (Y/N/U)
ALK Mutation (Y/N/U)
Post-op complication (Y/N)
Re-op (Y/N) if Y Reason
Adjuvant treatment (Y/N) (Chemotherapy, Radiation, Chemo-Radiation, Immunotherapy, Chemo Immunotherapy, Chemo-Radio-Immunotherapy, Radio-Immunotherapy)
- If Chemotherapy: type of Chemo

- If Radiotherapy: regimen
Recurrence (Y/N) if Y: date of recurrence
Death (Y/N) if Y: date of death

## Data management

Data for this study will be collected, recorded and stored using REDCap (Research Electronic Data Capture). REDCap is a secure, web application designed to support data capture for research studies. It includes features for HIPAA (Health Insurance Portability and Account-ability Act) compliance including real-time data entry validation (e.g. for data types and range checks), a full audit trail, user-based privileges, de-identified data export mechanism to statistical packages (SPSS), and integration with the institutional Active Directory. Access to study data in REDCap will be restricted to the members of the study team with authentication through hospitals credentials. The REDCap database and web server are housed on secure platforms that are backed up daily. The data collected for the review will be stored at Lucerne Cantonal Hospital under the direction of the Principal Investigator. In accordance with the Medical Professional Code, all important study documents (e.g., CRFs) will be archived for at least ten years after the end of the study.

The results of the study will be published in peer-reviewed journals and presented to the scientific community at national and international conferences.

## Statistical considerations

### Sample size calculation

The minimum sample size to estimate the incidence was calculated using the method described by Humphrey et al. [9]:

$$Sample\ size = (1.96Ie * L)2 * Ie * (1 - Ie)$$

Setting the expected incidence (of patients with an N1 lymph node metastasis to 0.2, and the allowable error of the incidence (L) to 0.2 the minimum required sample size is 385 patients. The allowable error was chosen because it represents a reasonable balance between precision and feasibility. After doubling the number to account for loss to follow-up and a lower true incidence, the final sample size is 770 patients.

The minimum sample size for the survival analysis was calculated using the method described by Schoenfeld [10]:

$$Events\ needed = \frac{(Z_\alpha + Z_\beta)^2}{(\log(RH)^2 * (1 - q_1) * q_1)}$$

We set the type I error rate to 0.05 (), and the type II error rate to 0.2 (). Based on previous studies the expected percentage of patients with N1 lymph node metastasis was set to 20% (q1) and the relative hazard (RH) was set to 1.5 [11–13]. The number of events needed to achieve a power of 80% in the cohort was 298. To account for this we increased the final sample size by 10% to 857.

### Statistical analysis

The incidence of N1 lymph node metastasis overall and the incidence of metastasis to the different lymph node stations (Hilar 10/11, Lobar 12, Sublobar 13/14) will be calculated by dividing the number of the respective events by the patient years separately. Ninety-five percent confidence intervals will be calculated using the method described by Ulm [14].

To investigate the association between visceral pleural invasion and the presence of metastatic lymph nodes univariable and multivariable logistic regression models will be fitted to the data. The multivariable logistic regression will be adjusted for age, sex, and tumor characteristics (size, histology, and mutation status).

For the secondary outcomes, Kaplan-Meier curves for the overall cohort and the subgroups (presence of N1 metastasis (Yes/No), presence of metastasis in intrapulmonary lymph nodes (Yes/No), presence of pleural invasion (Yes/No), sex (male/female), and age ($\geq$ median, < median) will be created, for the disease-free survival and overall survival separately. Invariable Cox proportional-hazards models will be fitted for each of the predictors of interest. Hazard ratios and corresponding 95% confidence intervals will be calculated from the coefficients. Furthermore, we will also create multivariable Cox models adjusting for age, sex, and relevant comorbidities. The assumption of constant hazards will be checked by calculating the scaled Schoenfeld residuals and inspecting the correlation with time both statistically and visually [10]. The assumption of non-informative censoring will be tested by creating optimistic models in which all censoring will be set to the longest survival time, and pessimistic models, in which all censored time points will be regarded as events. If the results point in the same direction, we will regard the censoring as non-informative. In case of a violation of the assumption of non-informative censoring we will conduct a thorough examination of key variables, such as disease status, surgery success, and follow-up length, to identify any relationship between these variables and the censoring event as well as the outcome of interest. If we find any evidence of such a relation, we will then adjust for the relevant variables and perform sensitivity analyses to evaluate the robustness of our findings to potential violations of the non-informative censoring assumption.

If the number of missing data of a variable does not exceed 5%, it will be considered for impuatation. The relationship of missingness between different variables will be graphically explored using the "naniar" package for R. Heatmaps of missing data and scatter plots of numerical data indicating missigness will be created. If the data is considered to be at "missing at random" or "missing completely at random" we will impute it using a random-forest based imputation algorithm as implemented in the "missForest" package for R. This algorithm has been shown to perform as good as other state-of-the-art imputation methods and can handle a wide array of data types [15]. All analysis will be performed using the statistical software "R" (Version 4.1.2).

## Ethical considerations

Individual patient consent is not required as data are collected under existing permissions (general consent) cancer registration in each jurisdiction.

Only members of the study team will have access to the information. The purpose of sharing data is to increase the ability of the investigators to analyze and translate data into a meaningful report validating their individual findings. A Data Transfer Agreement (namely a formal contract that documents which data are being transferred, the format of the data collection file and how the data can be used) will be signed between the principal investigator and the local investigators. The agreement protects all the centers who will provide the data, ensuring that data will not be misused. Each center is represented by one local investigator who will participate to regular meetings, held quarterly throughout the year, where the trend of the data collection along with critical points will be discussed in order to improve the understanding of the project and protocol.

Data will be de-identified, coded securely, and stored on a password-protected, virus protected, firewalled and encrypted server at Cantonal Hospital of Lucerne. The data from the

other centers will be entered in REDCAP by the local investigator through a link provided by the principal investigator. Since this protocol describes a no-intervention study, the only risk to the patient would involve a breach of privacy. This risk is minimized, however, since only indirect identifiers will be recorded at all points throughout the course of the study, ensuring that patient confidentiality is safely kept and the patients will only be recorded in the database by their participant identification code. The link between this code and the patient identity will be in hard copy only and will be stored in a locked location separate from the data collections forms. This code document will be destroyed upon completion of the study data extraction. The members of the study team will maintain professionalism at all times, ensuring the full protection of each patient's privacy.

## Discussion

This study seeks to improve the accuracy of staging and prognostication of patient undergoing pulmonary resection due to NSCLC. The decision to routinely examine the intraparenchymal lymph nodes, as part of systematic mediastinal lymphadenectomy during lung surgery, is not well established to date Vaghjiani and coll. reported in a retrospective analysis published in 2020 a percentage of 31% of patients with only intraparenchymal lymph nodes metastases on a cohort of 129 patients with occult node metastases [16]. Maeshima and colleagues reported an incidence of positive 13/14 lymph nodes of 21% [13]. That being so, new data are needed in order to avoid an inaccurate pathologic staging.

In addition, the detection of metastatic intraparenchymal lymph node in patients with NSCLC and visceral pleural invasion could possible explain the worse DFS and OS observed in this subgroup.

VPI, indeed, is a relevant prognostic factor in the treatment of early stage NSCLC and the optimal extent of the surgical resection for early-stage NSCLC patients with VPI is to date under debate. The addition of the type of resection in the collected data will provide important informations about OS and DSF in patients with VPI and different extent of resection.

## Supporting information

**S1 Checklist. SPIRIT 2013 checklist: Recommended items to address in a clinical trial protocol and related documents\*.**
(PDF)

**S1 File.**
(PDF)

## Author Contributions

**Conceptualization:** Fabrizio Minervini, Pietro Bertoglio, Henning Nilius.

**Methodology:** Fabrizio Minervini, Allen Li, Henning Nilius.

**Project administration:** Fabrizio Minervini, Peter Kestenholz.

**Writing – original draft:** Fabrizio Minervini.

**Writing – review & editing:** Fabrizio Minervini, Peter Kestenholz, Pietro Bertoglio, Allen Li, Henning Nilius.

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
