## [Decision Letter · Decision Letter 0]

7 Mar 2023

PONE-D-22-33926Role of intrapulmonary lymph nodes in patients with NSCLC and visceral pleural invasion. The VPI 1314 multicenter registry study protocolPLOS ONE

Dear Dr. Minervini,

Thank you for submitting your manuscript to PLOS ONE. After careful consideration, we feel that it has merit but does not fully meet PLOS ONE’s publication criteria as it currently stands. Therefore, we invite you to submit a revised version of the manuscript that addresses the points raised during the review process.

ACADEMIC EDITOR: Four experts have reviewed the manuscript. Please explain in more details about the reviewer’s concerns and revise your paper. 

We look forward to receiving your revised manuscript.

Kind regards,

Ming-Ching Lee

Academic Editor

PLOS ONE

Journal Requirements:

2. Please ensure that you refer to Figure 1 in your text as, if accepted, production will need this reference to link the reader to the figure.

4. We note that the original protocol that you have uploaded as a Supporting Information file contains an institutional logo. As this logo is likely copyrighted, we ask that you please remove it from this file and upload an updated version upon resubmission.

Reviewers' comments:

Reviewer's Responses to Questions

**Comments to the Author**

1. Does the manuscript provide a valid rationale for the proposed study, with clearly identified and justified research questions?

Reviewer #1: Yes

Reviewer #2: Yes

Reviewer #3: Yes

Reviewer #4: Yes

2. Is the protocol technically sound and planned in a manner that will lead to a meaningful outcome and allow testing the stated hypotheses?

Reviewer #1: Yes

Reviewer #2: Yes

Reviewer #3: Yes

Reviewer #4: Yes

3. Is the methodology feasible and described in sufficient detail to allow the work to be replicable?

Reviewer #1: Yes

Reviewer #2: Yes

Reviewer #3: Yes

Reviewer #4: Yes

4. Have the authors described where all data underlying the findings will be made available when the study is complete?

Reviewer #1: Yes

Reviewer #2: Yes

Reviewer #3: Yes

Reviewer #4: No

5. Is the manuscript presented in an intelligible fashion and written in standard English?

Reviewer #1: Yes

Reviewer #2: Yes

Reviewer #3: Yes

Reviewer #4: Yes

6. Review Comments to the Author

You may also provide optional suggestions and comments to authors that they might find helpful in planning their study.

Reviewer #1: This is a well-written study protocol to retrospectively investigate the incidence of intraoperative lymph node metastasis and the association between intraoperative lymph node metastasis and visceral pleural invasion. Based on the recently revealed significant study results regarding sublobar resection (JCOG0802, CALGB140503), the use of sublobar resection has been increasing and is expected to increase further. To determine the extent of resection (sublobar resection or lobectomy), the presence of lymph node metastasis (occult lymph node metastasis) would be essential. Therefore, the results of this study will be of great importance to all thoracic surgeons. Vaghjiani et al. investigated occult lymph node metastasis in patients with clinical stage I lung cancer and found one-third of occult metastases located in intrapulmonary lymph nodes (Vaghjiani et al. J Thorac Oncol 2020). The findings of Vaghjiani’s study would be related to this study, which should be described in the Introduction or Discussion.

Reviewer #2: I red with interest the proposed protocol. It is well written ad focused on an actual and interesting matter regaradin lung cancer staging.

however, I have some criticisms that need author clarification.

1) which kind of lymphadenectomy are you planning for this study? for the study purpose I suggesto to insert a radical mediastinal dissection with almost 3 mediastinal nodal station dissected.

2) why do you prefer a zip file for data collection? Maybe for multicentric studies a web based dataset software is more fit.

Reviewer #3: the main issue I find is that unlike N2-nodes or even 10/11 nodes; the 12-14 nodes represent mainly the quality that is yielded from the pathology. Then becomes the issue of heterogeneity; as most centrest under report 12-14 nodes - compared to 11 nodes. So that is an insight into possible major bias. Thus; for reasoning - keep intrapulmonary nodes independent and other N1 nodes as divided.

Reviewer #4: Sample size: the formula for the prevalence estimation is not readable. Need justification of the allowable error. What is the 5-year survival rate? Will the cohort have enough events (298) during the study time?

Statistical analysis:

Should be “univariable” and “multivariable” instead of “univariate” and “multivariate”.

Which covariates will be included in the multivariable logistic regression?

What subgroups will be created for the KM analysis?

What if the non-informative censoring assumption is violated?

Need justification to use the RF for imputation? Will this imputation be used for outcome measures or covariates? Imputation will bring bias and a good approach requires a careful evaluation of the missing data mechanism and methods for imputation.

7. PLOS authors have the option to publish the peer review history of their article (what does this mean?). If published, this will include your full peer review and any attached files.

Reviewer #1: No

Reviewer #2: No

Reviewer #3: **Yes: **Ilkka Ilonen

Reviewer #4: No

---

## [Author Response · Author response to Decision Letter 0]

14 Mar 2023

Response to Reviewer #1:

Comment 1: This is a well-written study protocol to retrospectively investigate the incidence of intraoperative lymph node metastasis and the association between intraoperative lymph node metastasis and visceral pleural invasion. Based on the recently revealed significant study results regarding sublobar resection (JCOG0802, CALGB140503), the use of sublobar resection has been increasing and is expected to increase further. To determine the extent of resection (sublobar resection or lobectomy), the presence of lymph node metastasis (occult lymph node metastasis) would be essential. Therefore, the results of this study will be of great importance to all thoracic surgeons. Vaghjiani et al. investigated occult lymph node metastasis in patients with clinical stage I lung cancer and found one-third of occult metastases located in intrapulmonary lymph nodes (Vaghjiani et al. J Thorac Oncol 2020). The findings of Vaghjiani’s study would be related to this study, which should be described in the Introduction or Discussion.

Reply 1: we thank the reviewer for this comment which helped us to improve the manuscript. We have added the citation in the discussion

Response to Reviewer #2: 

Comment 1: I red with interest the proposed protocol. It is well written ad focused on an actual and interesting matter regaradin lung cancer staging. however, I have some criticisms that need author clarification.

which kind of lymphadenectomy are you planning for this study? for the study purpose I suggesto to insert a radical mediastinal dissection with almost 3 mediastinal nodal station dissected.

Reply 1: dear Reviewer thanks for your comment. Indeed we added a sentence accepting your suggestion (“radical mediastinal lymphadenectomy with at least 6 lymph nodes from 3 mediastinal nodal stations”).

Comment 2: why do you prefer a zip file for data collection? Maybe for multicentric studies a web based dataset software is more fit.

Reply 2: thanks for this suggestion, we added a sentence in the section “ethical consideration”. We will use links to REDCAP Database provided by the principal investigator to the local investigators. 

Response to Reviewer #3: 

Comment 1: the main issue I find is that unlike N2-nodes or even 10/11 nodes; the 12-14 nodes represent mainly the quality that is yielded from the pathology. Then becomes the issue of heterogeneity; as most centres under report 12-14 nodes - compared to 11 nodes. So that is an insight into possible major bias. Thus; for reasoning - keep intrapulmonary nodes independent and other N1 nodes as divided.

Reply 1: dear Reviewer thanks a lot for your comment. In our real life scenario, the nodal stations 13/14 are often not analyzed and thus we would like to investigate if: 1- there is a percentage of patients which are nodal positive (stations 13/14) but they don’t receive adjuvant chemotherapy because the thoracic surgeons or the pathologists are not actively searching for the intraparenchymal nodes. 

2- the worse OS and DFS of patients with NSCLC and visceral pleural invasion could be attributed to an unknown N1 Status of the intraparenchymal nodal stations.

In order to minimize the bias connected with the quality of the pathology, two seniors (>10 years experience) pathologists will examine the lymph nodes for the study purposes. The surgeon will dissect all the mediastinal lymphnodes trough a radical mediastinal lymphadenectomy during surgery and the pathologists will search for the lymph nodes station 13/14.

Response to Reviewer #4: 

Comment 1: Sample size: the formula for the prevalence estimation is not readable. Need justification of the allowable error. What is the 5-year survival rate? Will the cohort have enough events (298) during the study time?

Reply 1: We thank the reviewer for raising this point. We re-inserted the formula and added a justification for the allowable error. According to the SEER database of American National Cancer Institute the 5-year survival for localized NSCLC is 61.2 % (https://seer.cancer.gov/statfacts/html/lungb.html). According to this number enough events should happen in the study time. However, we increased the minimal sample size by an additional 10% and added a sentence to the manuscript. 

Comment 2: Should be “univariable” and “multivariable” instead of “univariate” and “multivariate”.

Reply 2: We thank the reviewer for catching this mistake. We have changed it throughout the manuscript

Comment 3: Which covariates will be included in the multivariable logistic regression?

Reply 3: Thank you for pointing this out. We will adjust for age, sex, and tumor characteristics (size, histology, and mutation status). We have added a sentence to the manuscript.

Comment 4: What subgroups will be created for the KM analysis?

Reply 4: Subgroups will include: presence of N1 metastasis (Yes/No), presence of metastasis in intrapulmonary lymph nodes (Yes/No), presence of pleural invasion (Yes/No), sex (male/female), and age (≥ median, < median).

Comment 5: What if the non-informative censoring assumption is violated?

Reply 5: Thank you very much for giving us the opportunity to describe this more in detail. If we find evidence of informative censoring, we will thoroughly investigate if there are variables that are related both to the outcome and the censoring. In case such relation is present, we will then adjust for the relevant variables and perform sensitivity analyses to evaluate the robustness of our findings to potential violations of the non-informative censoring assumption. We have added a description of this to the manuscript

Comment 6: Need justification to use the RF for imputation? Will this imputation be used for outcome measures or covariates? Imputation will bring bias and a good approach requires a careful evaluation of the missing data mechanism and methods for imputation

Reply 6: Thank you very much for raising this important point. We fully agree that a good approach is required when performing data imputation. Therefore, we added a more detailed description of the process. Also, since we plan to only impute the data where less than 5% of the data are missing the bias introduced by the imputation should be relatively small. The random forest based imputation algorithm was chosen because it performs similar to other modern imputation methods and can handle a wide variety of data types.

---

## [Decision Letter · Decision Letter 1]

18 Apr 2023

Role of intrapulmonary lymph nodes in patients with NSCLC and visceral pleural invasion. The VPI 1314 multicenter registry study protocol

PONE-D-22-33926R1

Dear Dr. Minervini,

We’re pleased to inform you that your manuscript has been judged scientifically suitable for publication and will be formally accepted for publication once it meets all outstanding technical requirements.

Kind regards,

Ming-Ching Lee

Academic Editor

PLOS ONE

Additional Editor Comments (optional):

Reviewers' comments:

Reviewer's Responses to Questions

**Comments to the Author**

1. Does the manuscript provide a valid rationale for the proposed study, with clearly identified and justified research questions?

Reviewer #1: Yes

Reviewer #2: Yes

Reviewer #3: Yes

Reviewer #4: Yes

2. Is the protocol technically sound and planned in a manner that will lead to a meaningful outcome and allow testing the stated hypotheses?

Reviewer #1: Yes

Reviewer #2: Yes

Reviewer #3: Yes

Reviewer #4: Yes

3. Is the methodology feasible and described in sufficient detail to allow the work to be replicable?

Reviewer #1: Yes

Reviewer #2: Yes

Reviewer #3: Yes

Reviewer #4: Yes

4. Have the authors described where all data underlying the findings will be made available when the study is complete?

Reviewer #1: Yes

Reviewer #2: Yes

Reviewer #3: Yes

Reviewer #4: Yes

5. Is the manuscript presented in an intelligible fashion and written in standard English?

Reviewer #1: Yes

Reviewer #2: Yes

Reviewer #3: Yes

Reviewer #4: Yes

6. Review Comments to the Author

You may also provide optional suggestions and comments to authors that they might find helpful in planning their study.

Reviewer #1: I appreciate that the authors revised their paper appropriately. I do not have any further comments.

Reviewer #2: The authors answered clearly to my comments. I think this is an interesting protocol and I', courious to see their results.

Reviewer #3: all concerns are addressed satisfactorilrly. I wish all the best to the authors and to the project presented

Reviewer #4: All my concerns were addressed.

The statistical section is acceptable.

I think the protocol is good for publication.

7. PLOS authors have the option to publish the peer review history of their article (what does this mean?). If published, this will include your full peer review and any attached files.

Reviewer #1: No

Reviewer #2: No

Reviewer #3: No

Reviewer #4: No

---

## [Editor Report · Acceptance letter]

24 Apr 2023

PONE-D-22-33926R1 

Role of intrapulmonary lymph nodes in patients with NSCLC and visceral pleural invasion. The VPI 1314 multicenter registry study protocol 

Dear Dr. Minervini:

I'm pleased to inform you that your manuscript has been deemed suitable for publication in PLOS ONE. Congratulations! Your manuscript is now with our production department. 

Kind regards, 

on behalf of

Dr. Ming-Ching Lee 

Academic Editor

PLOS ONE